# A Low-Resource Training Strategy for Cell Segmentation using Patch-Based Attention U-Net

**Kai-Lin Chen**[1]                                                B11508002@NTU.EDU.TW

**Yu-Nong Lin**[1]                                                LINYUNONG@NTU.EDU.TW

**Pen-hsiu Grace Chao**[1]                                        PGCHAO@NTU.EDU.TW

**Kevin T. Chen**[1]                                              CHENKT@NTU.EDU.TW

[1] *Department of Biomedical Engineering, National Taiwan University, Taipei, Taiwan*

**Editors:** Accepted for publication at MIDL 2025

## Abstract

Segmentation is an essential tool for cell biologists and involves isolating cells or cellular features from microscopy images. An automated segmentation pipeline with high precision and accuracy can significantly reduce manual labor and subjectivity. Frequently, researchers would seek for a validated model available online and fine-tune it to meet their segmentation requirements. However, the established fine-tuning approach may involve online training or computationally intensive offline training. To address this, we propose an offline training pipeline requiring only tens of samples that are morphologically distinct from pre-training data. Specifically, we employed a patch-based attention U-Net trained with a threshold-based custom loss function. Finally, we evaluated this workflow along with two other state-of-the-art models, Stardist and Cellpose, on three different tasks. Our method improves image segmentation performance by 32.60% and 35.62% over Stardist and Cellpose, respectively, using the same amount of training samples. The code is available on our GitHub page: https://github.com/NTUMMIO/PAULow.

**Keywords:** Low-resource learning, Cell segmentation, Microscopy imaging, Custom loss

## 1. Introduction

Training a robust model for image segmentation can significantly shorten the time spent on image analysis. However, deep learning models would require a substantial amount of labeled data, often difficult and time-consuming to acquire (Román et al., 2023; Kemeter et al., 2024); without sufficient training data, the model may suffer from overfitting (Li et al., 2019; Ye et al., 2022). Recently, multiple deep learning-based cell microscopy image segmentation models trained on large datasets, such as Cellpose and Stardist (Schmidt et al., 2018; Stringer et al., 2020; Pachitariu and Stringer, 2022; Stringer and Pachitariu, 2025), are available online for cell biologists. Although these works allow users to fine-tune the models to a specific dataset, current fine-tuning strategies were mostly applicable to segmentation on whole cells with relatively round or regular morphology (Kleinberg et al., 2022; Liu et al., 2024). Moreover, successful fine-tuning often depends on the sample size and morphology of the target dataset (Tinn et al., 2021; Davila et al., 2024). To establish a low-resource training strategy adaptive to morphological variability in regions of interest (ROI), we design a unified training framework that integrates a patch-based cropping mechanism for normalization of input dimensions and augment data diversity. The proposed framework utilized an attention-based U-Net architecture and its generalizability and robustness was evaluated through a 5-fold cross-validation scheme across datasets with diverse ROIs.

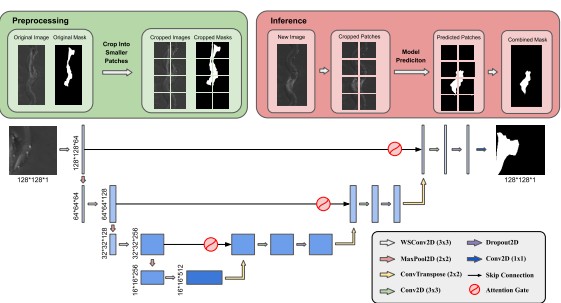

Figure 1: The overview of the proposed training method. Training images and masks were cropped to 128×128 to ensure consistent input tensor sizes. WSConv (Weight Standard Convolution) enhances Conv (Standard convolution) layers by normalizing weights to improve training stability. Our designed dynamic loss function enables the model to distinguish background-dominant and foreground-dominant patches.

## 2. Methodologies

### 2.1. Dataset Introduction and Preprocessing

In this study, we selected three datasets ranging from trivial to challenging, consisting of fluorescently labeled cell nuclei (Task 1), phase-contrast images of cells (Task 2), and fluorescently labeled actin filaments (Task 3), acquired on inverted Leica DMI3000B or Nikon Ti2E systems, equipped with an sCMOs camera (Hamamatsu ORCA-Flash4.0 LT), and a 20× objective (Wen et al., 2022). Ground truths are labelled by an experienced cell biologist. Each dataset has a different mean signal-to-noise ratio (SNR) value and image dimensions as presented in Table 1. Representative images from various ROIs are illustrated in Figure 2, each showing a different region of interest. The original image and mask were zero-padded such that their edges were multiples of 128 pixels and then cropped into 128×128 patches.

Table 1: Dataset Overview

| Task | Mean SNR Value | Image Dimensions | Training Samples | Test Samples |
|---|---|---|---|---|
| Task 1 | 28.9020 | $192 \times 436 \times 1$ | 21 | 10 |
| Task 2 | 10.4773 | $136 \times 558 \times 1$ | 21 | 10 |
| Task 3 | 2.8844 | $1024 \times 250 \times 1$ | 10 | 4 |

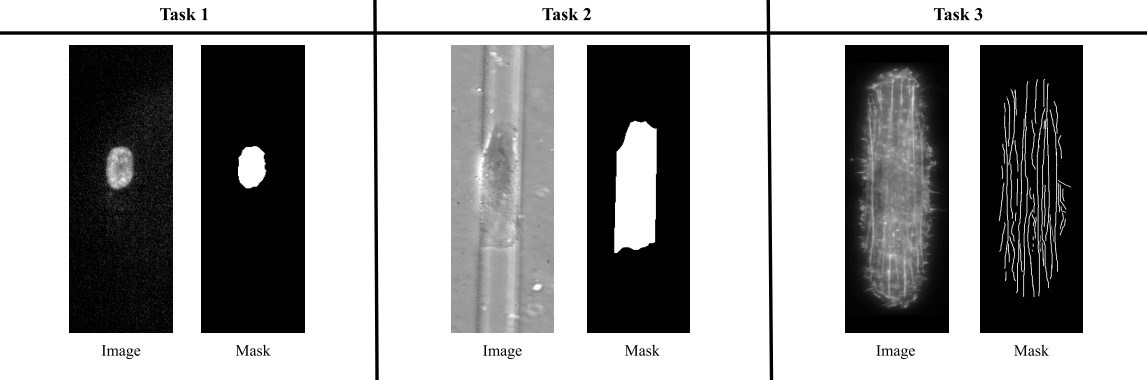

Figure 2: Sample images and masks from each task, each showing a different region of interest.

## 2.2. Network Design and Data-Efficient Training

Our network architecture is depicted in Figure 1. We adopted the Attention U-Net structure (Oktay et al., 2018), utilizing the gated attention mechanisms to focus on relevant regions. To maximize the utility of the dataset, we propose to apply a patch-based approach (Ullah et al., 2023) that enables precise pixel-wise segmentation and data augmentation. A threshold-based dynamic loss function was employed to address foreground-background imbalance in the cropped dataset (see below). In contrast, the benchmark models were fine-tuned following the optimal settings recommended by their respective authors.

## 2.3. Dynamic Threshold-based Loss Function Selection

To address the foreground-background imbalance in the cropped dataset while retaining all training patches, we proposed an adaptive, dynamic, threshold-based loss function strategy. Predicted masks were categorized into three types: background patches, small ROI patches, and large ROI patches. A patch was classified as a small ROI patch if the predicted foreground area was less than 6.25% of the patch. The overall loss framework combined three loss functions: Binary Cross-Entropy (BCE) loss, Tversky loss, and Focal Tversky loss. Background patches used only BCE loss to encourage true negative predictions. Small ROI patches were optimized using a combination of BCE and Tversky loss to penalize false positives while reinforcing true negatives. In contrast, large ROI patches used a combination of Tversky loss and Focal Tversky loss to intensify penalties on both false positives and false negatives, promoting pixel-wise segmentation in the training patches.

Table 2: Model Segmentation Performance (Dice Score) Across Three Tasks

| Model | Task 1 | Task 2 | Task 3 | Average | Task Coverage | Final Score |
|---|---|---|---|---|---|---|
| Stardist | 0.8232 | 0.5904 | 0.5164 | 0.6433 | 1.0000 | 0.6433 |
| Cellpose | 0.9834 | 0.9036 | N/A | 0.9435 | 0.6667 | 0.6290 |
| Proposed Method | 0.9908 | 0.9327 | 0.6357 | 0.8531 | 1.0000 | 0.8531 |

## 3. Results and Conclusions

Segmentation performance was evaluated using the mean Dice score across three tasks, excluding shape-incompatible cases, with scores weighted by task coverage for fairness. As shown in Table 2, our method outperformed Cellpose and Stardist across all three tasks, achieving a score of 0.8531(Stardist = 0.6433 and Cellpose = 0.6290) without the need for large-scale pretraining. Using the same number of training samples, our patch-based attention training network showed improvements in terms of final score by 32.60% and 35.62% compared to Stardist and Cellpose, respectively. Overall, our proposed method achieved a higher final score which suggested its greater adaptability to diverse target morphologies as compared to benchmark models. In conclusion, this study demonstrated that the proposed patch-based attention mechanism can serve as a lightweight and adaptable alternative among existing cell image segmentation models.

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
