# OpenReview forum: "A Low-Resource Training Strategy for Cell Segmentation using Patch-Based Attention U-Net"
_MIDL.io/2025/Short_Papers — MIDL 2025 - Short Papers_

### Official Review · Reviewer_z3tg · 2025-04-28

**Rating:** 2
**Confidence:** 4

**Summary:**

This paper adopts an Attention U-Net with patch sampling to enable few-shot learning for cell segmentation. Extensive experiments on three datasets demonstrate the proposed pipeline's effectiveness over existing methods, i.e., Stardist and Cellpose.

**Strengths:**

•	The paper is well-written, and the authors have made their code publicly available.
•	The experiments are conducted on three datasets, lending credibility to the conclusions.

**Weaknesses:**

•	The methodology is relatively conventional. Patch-based strategies are widely used in medical image analysis, and the Attention U-Net is an older backbone. It would be more compelling to explore more recent lightweight models, such as transformer- or Mamba-based architectures.
•	The threshold-based loss function is only briefly mentioned without a clear introduction, leaving readers unclear about how it differs from standard loss functions.

---

### Decision · Program_Chairs · 2025-05-01

**Decision:**

Accept

**Comment:**

The PC discussed the paper during the panel meeting and decided to accept. Given the scope of the short paper track, the PC believes this paper provides a sufficient contribution to warrant presentation at MIDL.